A rapid spread of the stony coral tissue loss disease outbreak in the Mexican Caribbean

http://orcid.org/0000-0002-5726-7238 Alvarez-Filip Lorenzo lorenzoaf@gmail.com
Estrada-Saldívar Nuria
Pérez-Cervantes Esmeralda
Molina-Hernández Ana
http://orcid.org/0000-0001-5089-0687 González-Barrios Francisco J.
Biodiversity and Reef Conservation Laboratory, Unidad Académica de Sistemas Arrecifales, Instituto de Ciencias del Mar y Limnología, Universidad Nacional Autónoma de México , Puerto Morelos, Quintana Roo , Mexico
Nelson Craig
Electronic publication date: 2019 Nov 26
Publication date: 2019
Volume: 7
Electronic Location ID: e8069
Received 2019 Aug 12; Accepted 2019 Oct 20
Copyright: © 2019 Alvarez-Filip et al.
Copyright year: 2019
Copyright holder: Alvarez-Filip et al.
License: This is an open access article distributed under the terms of the Creative Commons Attribution License, which permits unrestricted use, distribution, reproduction and adaptation in any medium and for any purpose provided that it is properly attributed. For attribution, the original author(s), title, publication source (PeerJ) and either DOI or URL of the article must be cited.
License URL: https://creativecommons.org/licenses/by/4.0/

Keywords: White plague; Coral mortality; Disease prevalence; Reef monitoring; Long-term data, Reef functioning; White syndrome; SCTLD

Funding: Consejo Nacional de Ciencia y Tecnologia CONACyT PDC-247104 Royal Society Newton Advanced Fellowship NA150360 Universidad Nacional Autónoma de México (Program UNAM-DGAPA-PAPIIT) IN-205019 Healthy Reefs Initiative and the Comision Nacional de Áreas Naturales Protegidas This study was supported by the Consejo Nacional de Ciencia y Tecnologia (CONACyT; PDC-247104), a Royal Society Newton Advanced Fellowship (NA150360), the Universidad Nacional Autónoma de México (Program UNAM-DGAPA-PAPIIT, project IN-205019), and the Healthy Reefs Initiative and the Comision Nacional de Áreas Naturales Protegidas. The funders had no role in study design, data collection and analysis, decision to publish, or preparation of the manuscript.

==============================
Caribbean reef corals have experienced unprecedented declines from climate change, anthropogenic stressors and infectious diseases in recent decades. Since 2014, a highly lethal, new disease, called stony coral tissue loss disease, has impacted many reef-coral species in Florida. During the summer of 2018, we noticed an anomalously high disease prevalence affecting different coral species in the northern portion of the Mexican Caribbean. We assessed the severity of this outbreak in 2018/2019 using the AGRRA coral protocol to survey 82 reef sites across the Mexican Caribbean. Then, using a subset of 14 sites, we detailed information from before the outbreak (2016/2017) to explore the consequences of the disease on the condition and composition of coral communities. Our findings show that the disease outbreak has already spread across the entire region by affecting similar species (with similar disease patterns) to those previously described for Florida. However, we observed a great variability in prevalence and tissue mortality that was not attributable to any geographical gradient. Using long-term data, we determined that there is no evidence of such high coral disease prevalence anywhere in the region before 2018, which suggests that the entire Mexican Caribbean was afflicted by the disease within a few months. The analysis of sites that contained pre-outbreak information showed that this event considerably increased coral mortality and severely changed the structure of coral communities in the region. Given the high prevalence and lethality of this disease, and the high number of susceptible species, we encourage reef researchers, managers and stakeholders across the Western Atlantic to accord it the highest priority for the near future.

Introduction

Over the past four decades, coral reefs have experienced declines in their condition and function, which has been attributed to coral disease, overfishing and herbivore loss, eutrophication, sedimentation, and climate change (Jackson et al., 2014; Hughes et al., 2017; Lapointe et al., 2019). For the Caribbean in particular, diseases have caused devastating declines in living coral cover of more than 50–80% within a few decades (Aronson & Precht, 2001; Jackson et al., 2014). Furthermore, diseases have also impacted populations of other key components of Caribbean ecosystems, such as the decimation of Diadema antillarum populations due to a non-identified pathogen during the 1980s (Lessios, Robertson & Cubit, 1984). The region-wide outbreak of the white-band disease in the late 1970s led to a substantial loss of the major reef-building corals Acropora palmata and Acropora cervicornis (Gladfelter, 1982; Aronson & Precht, 2001). It is estimated that nearly 80% of the population was lost during this event (Gladfelter, 1982; Aronson & Precht, 1997). In the late 1990s, the white-pox disease, apparently caused by a human-related pathogen, further decimated the populations of Acropora palmata (Patterson et al., 2002). In addition, the increase in the incidence of the yellow-band disease during the 1990s affected the populations of Orbicella spp., which are important reef-building species that tend to dominate many fore-reef zones (Cervino et al., 2001; Gil-Agudelo et al., 2004). Multiple events of white-plague disease outbreaks during the last decades have also substantially decimated the populations of a range of coral species (Weil, 2004; Harvell et al., 2007; Precht et al., 2016). It has been suggested for some sites that white-plague disease may have a greater impact on the Caribbean than other diseases (Cróquer et al., 2005). Due to the fact that the most severely impacted coral species are also major reef-building corals, disease outbreaks in the Caribbean have largely contributed to the substantial changes in spatial heterogeneity and ecological functionality of Caribbean reefs, along with their capacity to provide important ecosystem services to humans (Alvarez-Filip et al., 2009; Aronson & Precht, 2001; Weil, 2004).

Although Caribbean reef-related diseases were first reported in the early 1970s, our knowledge of their pathology, etiology, and epizootiology (i.e., what are the main drivers that potentially trigger a disease outbreaks) of most coral reef diseases is still limited. However, it is likely that increasing pressures in the form of climate change and coastal development will increase disease prevalence and the negative effects of diseases on coral communities. For instance, coral diseases are likely to be exacerbated in a context of rapidly increasing sea surface temperatures, as thermal stress has been linked to coral disease outbreaks (Bruno et al., 2007; Randall et al., 2014; van Woesik & McCaffrey, 2017). In addition, coral diseases have also been related to stressors such as excess nutrients from sewage or high levels of sedimentation (Sutherland et al., 2010; Bruno et al., 2003).

In 2014, a new emergent coral disease, the stony coral tissue loss disease (SCTLD), was first reported off the coast of Miami-Dade County, Florida in September 2014, just after an intense bleaching event during the summer of the same year (Precht et al., 2016; Precht, 2019; Florida Department Environmental Protection (FDEP), 2019). Since then, the SCTLD has gradually spread across the Florida Reef Tract (Florida Department Environmental Protection (FDEP), 2019) and began to reach other regions in the Caribbean (AGRRA, 2019). In Florida, regional declines in coral density approached 30% loss and live tissue loss was upward of 60% as a result of the disease outbreak (Walton, Hayes & Gilliam, 2018). The cause of the disease is still unknown but it is affecting more than 20 species of corals (Florida Department Environmental Protection (FDEP), 2019), with highly-susceptible species showing initial signs of infection, followed by intermediate-susceptible species (Florida Department Environmental Protection (FDEP), 2019). The most evident symptom is the display of multiple lesions that provoke rapid tissue loss, leading to the exposure of bright white skeletons that are rapidly covered by turf, macroalgae or sediment. Highly susceptible species include Pseudodiploria strigosa, Dendrogyra cylindrus, Meandrina meandrites, Dichocoenia stokesii, Montastraea cavernosa and Eusmilia fastigiata, among others (Precht et al., 2016; Precht, 2019; Florida Department Environmental Protection (FDEP), 2019). According to early reports, the SCTLD has not shown seasonal patterns linked to warming or cooling ocean temperatures, contrary to previous white plague diseases that have subsided in winter months as temperatures cooled (Harding et al., 2008; Miller et al., 2009; Florida Department Environmental Protection (FDEP), 2019).

On July 2018, following alerting reports issued by local divers, and in collaboration with the authorities of the Parque Nacional Arrecife de Puerto Morelos, we found a reef near Puerto Morelos, in the northern Mexican Caribbean that had a severe outbreak of a coral disease affecting similar species and exhibiting similar patterns as those previously reported in Florida (Fig. 1). Since then, we set out to survey other reefs in the Mexican Caribbean and found that the disease outbreak spread quickly across the region. Here we document the impact of the SCTLD on coral communities in the Mexican Caribbean by (i) quantifying the disease prevalence at 82 sites, and (ii) describing how this disease has modified the condition and composition of coral communities at 14 sites by using detailed information from before the onset of the outbreak.

Figure 1 Susceptible species affected by Stony Coral Tissue Loss Disease during the outbreak.

(A) Two colonies of Pseudodiploria strigosa observed on July 3, 2018 at a fore-reef reef site in Puerto Morelos, Mexico. One colony (front) shows the classic symptoms of the Stony Coral Tissue Loss Disease, while the other one died shortly before the photo was taken (recent and transient mortality). A Foureye butterflyfish (Chaetodon capistratus) is feeding on the edge of the lesion on the colony at the front. (B) Diseased colony of Montastraea cavernosa in direct contact with an apparently healthy colony of the same species observed in Cozumel on July 3, 2019. The close-up shows the touching-boundaries between the two colonies. Photo credits: Lorenzo Álvarez-Filip.

Materials and Methods

Data for this region-wide assessment was produced by the Healthy Reefs Initiative (HRI), the Comisión Nacional de Áreas Naturales Protegidas (Mexican Commission for Protected Areas; CONANP) and the Biodiversity and Reef Conservation Laboratory, UNAM. Sites were defined according to ongoing monitoring programs and research projects, therefore it was not possible to design a survey protocol to systematically represent reef gradients or zonation. A total of 82 sites were surveyed over the period July 2018–April 2019 (Table S1). A total of 77 sites are fore-reefs (7–15 m deep) and five are back-reefs (1–3 m deep). All sites were surveyed using the AGRRA coral protocol (Lang et al., 2011).

At each site, coral communities were surveyed by replicating two to 16 (mean = 4.3; SD = 3.7) belt transects of 10 × 1 m (Table S1). Although two transects is the minimum recommended by the AGRRA coral protocol (Lang et al., 2011), we aimed for more transects when possible in order to have a better representation of the coral community and disease prevalence. The following data were recorded for each coral colony within the transect: species name, colony size (maximum diameter, diameter perpendicular to the maximum and height), percentage of bleaching, percentage of mortality (new, transition and old) and the presence of SCTLD and other diseases (Lang et al., 2011). We then calculated the SCTLD prevalence at each site and for all coral species. For this study, we also recorded colonies with 100% mortality for which death could be attributable to the SCTLD (i.e., recent or transient mortality was still evident; see Fig. 1). To provide a clearer picture of the magnitude of the problem, we focused on exploring geographical and temporal trends for the 11 most “highly susceptible species,” which we defined as those that presented more than 10% of SCTLD prevalence across all surveyed sites (Fig. 2; Table S2).

Figure 2 Prevalence of the Stony Coral Tissue Loss Disease for the 11 most susceptible species across 82 reef sites in the Mexican Caribbean (n = number of colonies).

For this figure, we include coral colonies with total mortality but for which death could be attributable to the SCTLD (exposed bright white skeletons; see Fig. 1).

To identify whether the SCTLD outbreak may have started earlier than the summer of 2018, a variety of published and unpublished sources were used to provide a yearly estimate of disease prevalence at a regional level. Datasets were obtained from AGRRA, the HRI, CONANP monitoring protocols and scientific sources (publications and researchers), and are being systematized in the Coral Reef Information System of the Biodiversity and Reef Conservation Laboratory, UNAM (Table S1). Since the main intention of this exercise was to provide a regional perspective of disease prevalence, we only used years for which enough geographical representation exists. In other words, we included years with information from at least 17 sites distributed in at least three of the main sub-regions identified for the Mexican Caribbean (Northern Quintana Roo, Central Quintana Roo, Southern Quintana Roo, Cozumel and Chinchorro Bank; Rioja-Nieto & Álvarez-Filip, 2019). In total, we present data for seven time periods: 2005/2006, 2009, 2011/2012, 2014, 2016, 2017 and 2018/2019. Some years were combined into one period, as they were part of the same monitoring campaign (i.e., sites were surveyed only once within each period).

In 2016 and 2017 we conducted an extensive effort to survey coral reefs systems across the Mexican Caribbean (Suchley & Alvarez-Filip, 2018; Perry et al., 2018). Although surveying the condition of coral communities was not part of the objectives of those campaigns, we assessed coral communities in 14 of those sites using the AGRRA methodology (see above). In 2018 and 2019 we revisited these 14 sites to compare how coral condition and coral community composition changed from before the SCTLD outbreak to after the onset of the outbreak (in 2018/2019). These sites are distributed in the northern portion of the Mexican Caribbean: from Punta Allen to Puerto Morelos in the mainland, and in the windward and leeward coasts of Cozumel (Fig. 3). Eleven sites are fore-reefs and three are back-reefs. To describe patterns of coral mortality between periods, we first calculated the proportion of healthy, afflicted and dead colonies for each period (2016/2017 and 2018/2019). As described above, for this analysis we only considered the 11 most “highly susceptible species.”

Figure 3 Prevalence of the Stony Coral Tissue Loss Disease in the Mexican Caribbean.

Dots represent the location of the 82 surveyed reefs and the colors represent the SCTLD prevalence for the 11 most afflicted species (see methods and Fig. 2). Data on this figure was collected by the Healthy Reefs Initiative, the Comisión Nacional de Áreas Naturales Protegidas (Mexican Commission for Protected Areas; CONANP) and the Biodiversity and Reef Conservation Laboratory, UNAM. Please note that reef sites were surveyed at different times (between July 2018 and April 2019).

The variation in the overall coral community composition (including all recorded species) between 2016/2017 and 2018/2019 was investigated with non-metric multidimensional scaling (nMDS) based on Bray–Curtis similarities of square root transformed coral cover species data in Primer v6 (Clarke & Gorley, 2006). The matrix was created using the relative abundance of each healthy, afflicted, and dead colony for each coral species for each period. The relative abundance of each coral species was used as the variable, the sites as the samples. We used the before and after periods, and the reef zones as factors. A two-way Analysis of Similarities (ANOSIM) was used to test the significance of these groupings (9,999 permutations).

We then used the convex hull (polygon delineated by the exterior points of each period in the nMDS) to represent the variability of the coral community for each time period (2016–2017 vs 2018–2019), and the standard ellipse area (SEA) to quantitatively explore the overlap between the two periods. Briefly, the standard ellipse is to bivariate data as standard deviation is to univariate data. The standard ellipse of a set of bivariate data is calculated from the variance and covariance of the two axes and contains approximately 40% of the data (Jackson et al., 2012). To compare the total area for each time period, we used the Bayesian SEA corrected for the sample size (SEAc) estimated, and plotted it using the SIBER routine for the SIAR package in R (Parnell & Jackson, 2015). The overlap of the time period was calculated as the overlapping proportion of the SEAc (Jackson et al., 2011).

Results and Discussion

Here, we describe how the SCTLD affected 82 reef sites distributed along the Mexican Caribbean coast. More than 40% of the sites had a SCTLD prevalence of 10% or more, and nearly a quarter of the sites had a disease prevalence of more than 30% (Fig. 3). Our results should be taken as a conservative value, since many sites were surveyed when the SCTLD outbreak was only starting (i.e., only a few colonies of a few species were afflicted by the disease; Fig. 3). We observed great variability in prevalence that was not attributable to any geographical gradient or seasonality (Fig. 3). For example, the SCTLD was first observed in Cozumel’s windward coast in November 2018 (as in most of the surveyed sites in the mainland); however, it was not until December 2018 that the disease reached the reefs in the leeward side of Cozumel, followed by rapid spreading during the winter. Overall, the presence of the SCTLD in the Mexican Caribbean during 2018/2019 was well above the 5% disease prevalence, that has been identified as habitual for Caribbean reefs (Weil, 2004; Ruiz-Moreno et al., 2012), and just slightly lower than what has been reported for Florida a few years after the start of the SCTLD outbreak (Walton, Hayes & Gilliam, 2018). During the past 13 years, disease prevalence in the Mexican Caribbean was below 10%, and it reached its lowest point during 2016–2017, just 1 year before the SCTLD outbreak in this region, with only 1% prevalence (Fig. 4). Similarly throughout the Florida Reef Tract, the prevalence of disease before the first SCTLD reports was below 2%, but this prevalence doubled after the region-wide outbreak (Walton, Hayes & Gilliam, 2018). For this study, we present the temporal trend of disease prevalence to provide context to the SCTLD outbreak. However, we acknowledge that the information presented in Fig. 4 should be taken conservatively, since prevalence values are usually driven by few (sometimes different) afflicted species and are dependent on coral mortality from previous years.

Figure 4 Disease prevalence of the 11 most susceptible species to the Stony Coral Tissue Loss Disease (STCLD) from 2005/2006 to 2018/2019 in the Mexican Caribbean.

From 2009 to 2014 black-band disease was the most abundant coral disease and was mainly recorded in Siderastrea siderea in Cozumel.

For Florida, it has been shown that the 2014 SCTLD disease outbreak was linked to a severe thermal stress resulting from warm winter and spring temperatures followed by an anomalously warm summer (Gintert et al., 2019; van Woesik & McCaffrey, 2017; Walton, Hayes & Gilliam, 2018). For the Mexican Caribbean, we did not find evidence of intense thermal stress during this study (July 2018–April 2019). Coral bleaching was very low and unrelated to the disease prevalence in all our surveyed sites (Fig. S1). However, and although sea-surface temperatures rarely exceeded the bleaching threshold, temperature anomalies during the winter were relatively high (Cerdeira-Estrada et al., 2019; Fig. S1). The above-normal temperatures during the winter might have contributed to the rapid spread of the disease observed in Cozumel’s leeward coast during this season, since coral diseases are likely to be exacerbated by thermal stress (Bruno et al., 2007; Randall et al., 2014; van Woesik & McCaffrey, 2017). During this study, there were no other severe natural events, such as tropical storms or extraordinary rainfall that affected our study sites.

The SCTLD has a very high host range Our field surveys revealed that 24 out of 46 recorded species presented symptoms of SCTLD with a high disease prevalence: Dendrogyra cylindrus (71%, five out of seven colonies), Pseudodiploria strigosa (40%), Meandrina meandrites (38%), Eusmilia fastigiata (33%), Siderastrea siderea (26%), Diploria labyrinthiformis (25%), among others (Fig. 2, Table S2). As in Florida, we have also observed that some of the most susceptible species have disappeared from long-term monitoring sites. Potentially, this emergent disease has even driven local-extinction events of species such as Meandrina meandrites and Dendrogyra cylindrus, since these species that have vanished from several reef sites on the mainland coast of our study region. In fact, a recent study suggests that Dendrogyra cylindrus, a species that has been rare for hundreds of thousands of years, has a high likelihood of becoming extinct in the coming years due to this disease outbreak (Chan et al., 2019). We still recorded healthy colonies of Meandrina meandrites and Dendrogyra cylindrus at Chinchorro Bank and Cozumel Island, but some of the more recent surveys (not included in this study) revealed that the colonies from these two species are increasingly being afflicted by the SCTLD in Cozumel. Although there is the precedent of other diseases affecting multiple species in the Caribbean, such as the black band disease, yellow band syndrome and White Plague type II (Weil, 2004), it is possible that the SCTLD may become one of most lethal diseases in the recent history of the region, given the high levels of prevalence and high mortality rates on several coral species (see also Florida Department Environmental Protection (FDEP), 2019; Gintert et al., 2019).

Although there are some differences between the lists (and ranking) of afflicted species identified for the Mexican Caribbean (this study) and those reported for Florida (Precht et al., 2016; Walton, Hayes & Gilliam, 2018), the overall pattern is similar. Many of the species that remained as important reef-building corals after the declines of Acropora and Orbicella, are being severely affected by the SCTLD (González-Barrios & Álvarez-Filip, 2018; Fig. 2). Complexity-contributing species that exhibited significant declines include Pseudodiploria strigosa, Diploria labyrinthiformis, Colpophyllia natans and Montastraea cavernosa (Fig. 2). In contrast, we found very low disease prevalence on non-framework building corals such as Agaricia agaricites and Porites astreoides, which are species that have been previously described as intermediately-susceptible species to the SCTLD (Florida Department Environmental Protection (FDEP), 2019). These two species are very abundant across reef sites in the Mexican Caribbean and the wider Caribbean (Table S1; Green, Edmunds & Carpenter, 2008; González-Barrios & Álvarez-Filip, 2018). Therefore, the decline in the abundance of several species due to the SCTLD is likely to further increase the dominance of Agaricia agaricites and Porites astreoides in the region. This may have started to become apparent already. The relative abundance of these two species represented 46% of the surveyed colonies in 2016 and 2017, yet by 2018/2019 they accounted for 52% of the total number of recorded coral colonies. However, this is a preliminary observation and further studies should assess coral communities once the SCTLD has already passed its peak in the region. Overall, these findings suggest that the ultimate consequence of the SCTLD outbreak may be a further decrease on the physical persistence and ecological functionality of coral reefs (Alvarez-Filip et al., 2013; Perry et al., 2015; Perry & Alvarez-Filip, 2019).

Transmission, or the spread of disease among individuals, is a key factor in understanding the epidemiology and ecology of infectious diseases (Shore & Caldwell, 2019). Although the data for this study was not collected in a way that allowed us to quantitatively explore how the disease is transmitted between colonies, it is important to mention that we did not observe evidence that suggests a single potential mechanism of disease transmission. For example, we repeatedly observed, at different sites, individuals of the butterflyfish Chaetodon capistratus feeding on the edge of the lesions and then swimming away to feed off on other (healthy or afflicted) colonies (Fig. 1A). This behavior of Chaetodon capistratus was commonly observed on colonies of Pseudodiploria strigosa but occasionally on other susceptible species such as Diploria labyrinthiformis and Orbicella spp. During our field surveys it was also common to observe that neighboring colonies were afflicted (Fig. 1A), but in other instances we found colonies of the same species in direct contact with each other with no evidence of transmission from the afflicted colony to the apparently healthy colony (Fig. 1B). Thus, although it is suspected that the SCTLD is infectious (transmissible from a sick coral to a healthy coral), further pathological, etiological and genetic analyses are required to fully understand the sources and mechanisms of transmission of this emergent threat to Caribbean reefs.

The analysis of sites that contained pre-outbreak information showed that this outbreak event considerably increased coral mortality and severely changed the structure of coral communities in the region. In total, we surveyed 3,059 coral colonies of the highly susceptible species for both periods (2016/2017 and 2018/2019). During the pre-outbreak period, 99.5% of the coral colonies were healthy, yet during 2018–2019 the disease prevalence reached 25.9%, while another 12.9% of the colonies were already dead; probably as a consequence of the SCTLD (Fig. 5). All the colonies exhibited similar symptoms to those colonies from Florida, with rapid tissue loss occurring within a period of just a few weeks in the most extreme cases, leaving the white skeletons exposed, which were either colonized by macroalgae or covered by sediment shortly after. Additionally, our percentage of afflicted colonies by the SCTLD is similar to what was observed in Florida during 2014–2015, where they registered a 30% proportion of afflicted colonies (Precht et al., 2016). The coral community composition of those 14 sites changed considerably between the pre-outbreak surveys and 2018/2019 (Fig. 6). The two-way ANOSIM showed significant differences between sampling periods (R = 0.561, p = 0.001) and reef zones (R = 0.44, p = 0.004). The analysis revealed that the period of 2018/2019 had no overlap with the period of 2017/2019, which means that this corresponds to compositional changes in the coral reef communities. This is particularly explained by the sudden increase of afflicted colonies and of the number of dead colonies, especially from the species Meandrina meandrites, Pseudodiploria strigosa, Diploria labyrinthiformis and E. fastigiata. This massive disease-outbreak is a clear example of how coral diseases are drivers of change for coral communities (Harvell et al., 2007).

Figure 5 Proportion of healthy, afflicted and dead colonies of the highly susceptible species in 2016/2017, before the onset of the Stony Coral Tissue Loss Disease Outbreak (SCTLD) in the Mexican Caribbean, and in 2018/2019 when the SCTLD was spread across many sites in the region.

Figure 6 Coral community composition for the study sites before and after the disease.

Non-metric multi-dimensional scaling (nMDS) analysis displaying degree of similarity of the community composition across 14 sites in the Mexican Caribbean for the coral cover by species. The blue color represent the sites before the disease (2016–2017) and the gray color represent the sites after the disease (2018–2019). The circles represent the back-reef sites and the triangles the fore-reef sites. Dotted lines: convex hull total area (TA). Solid lines: standard ellipse area corrected for small sample sizes (SEAc).

The SCTLD outbreak reached the north of the Mesoamerican Reef System in 2018, which affected most of the coral reefs throughout the 400 linear km of the Mexican Caribbean coast in less than a year with a non-recognizable geographical pattern. This pattern contrasts the gradual spread observed across the Florida Reef Tract between 2014 and 2019 (Florida Department Environmental Protection (FDEP), 2019). The extremely rapid geographical progression of the SCTLD across the Mexican Caribbean could be explained, at least in part, by the rapidly decreasing quality of sea water in the region. The Mexican Caribbean coast has experienced dramatic coastal development over the last decades. Over 10 million tourists visit the region annually and the local population has grown exponentially (Suchley & Alvarez-Filip, 2018). Consequently, the coastal waters of the region have experienced eutrophication and increased sedimentation levels (Murray, 2007; Baker, Rodríguez-Martínez & Fogel, 2013; Hernández-Terrones et al., 2015). Eutrophication resulting from inadequate wastewater treatment has been previously identified as a major driver of declining reef condition in the region (Suchley & Alvarez-Filip, 2018). In addition and more recently, the Mexican Caribbean coast has regularly experienced a massive influx of drifting Sargassum that accumulates on the shores and rapidly decomposes, resulting in near-shore, murky-brown waters that rapidly increases nutrient concentration in the water column and reduces light, oxygen and pH levels (Van Tussenbroek et al., 2017). These sargassum-brown-tides have been proven to have drastic consequences on near-shore seagrass meadows and coral communities (Van Tussenbroek et al., 2017). Given the large amount of Sargassum reaching the coast, these negative effects are likely to disseminate further offshore reaching coral reefs (usually located 0.5–3 km from the coast). Further research is needed to fully comprehend the relationship between rapidly changing water quality in the Mexican Caribbean and the susceptibility of reef corals to diseases; however, chronic nutrient enrichment has already been related to coral diseases and bleaching under experimental conditions (Vega Thurber et al., 2014).

Conclusion

The Caribbean region is a well-known “disease hot-spot” because of the fast emergence, high prevalence, and virulence of coral-reef diseases and syndromes (Weil, 2004). These events have deeply marked the community composition of Caribbean reefs by decimating populations of important reef-building coral species such as Acropora palmata, which has not fully recovered from these events (Rodríguez-Martínez et al., 2014). However, the SCTLD is likely to become the most lethal coral disease ever recorded because of its high prevalence, the high number of susceptible species, its transmissibility, and the high levels of mortality exhibited by affected corals (Precht et al., 2016, 2018; Gintert et al., 2019). A total of 29 species, including rare and important reef-building coral species, have been reported to be affected by the SCTLD (this study; Precht et al., 2016; Walton, Hayes & Gilliam, 2018; Florida Department Environmental Protection (FDEP), 2019), but even more concerning is the fact that this disease is covering a wide geographic range and it is expanding rapidly. Recently, reports of the SCTLD have also been issued for Jamaica, St. Maarten, the Dominican Republic and St. Thomas in the U.S. Virgin Islands (AGRRA, 2019). The ultimate consequences for the wider Caribbean are yet to be seen; however, our findings suggest that this event has the potential to further decrease physical persistence and ecological functionality of coral reefs at a regional scale (Perry & Alvarez-Filip, 2019). Amelioration or eradication intervention have only partially succeeded in impeding the spread of the SCTLD disease across Florida and Mexico, in part because the disease is spreading more rapidly (weeks) than our capacity (scientists, managers, stakeholders) to respond to these types of events (Precht, 2019). Given the high prevalence and lethality of this disease, and the high number of susceptible species, we encourage reef researchers, managers and stakeholders across the Caribbean to accord it the highest priority for the near future.

Supplemental Information

Supplemental Information 1 Coral bleaching and thermal stress on the Mexican Caribbean between June 2018 and May 2019.

(A) Percentage of bleached colonies across the 82 surveyed reef sites (ordered by surveyed date). (B) Relationship between the percentage of bleached colonies and the stony coral tissue loss disease prevalence for the 82 surveyed sites (y = 0.43 x + 6.58, R² = 0.01). (C) Average monthly nighttime Sea Surface Temperature (MNSST) in blue (± St. Dev) and average monthly nighttime Sea Surface Temperature Anomaly (MNSST-A) in red (± St. Dev) from nine localities across the region (Cancun, Puerto Morelos, Cozumel East coast, Cozumel West coast, Tulum, Punta Allen, Mahahual, Xcalak and Chinchorro Bank). For (A) and (B) coral colonies completely and partially bleached were considered. For (C) the information was obtained from the Marine-Coastal Information and Analysis System of CONABIO (https://simar.conabio.gob.mx/), which use data from the following sources: Group for High Resolution Sea Surface Temperature (GHRSST), Jet Propulsion Laboratory (JPL), Physical Oceanography Distributed Active Archive Center (PO.DAAC), National Aeronautics and Space Administration (NASA). References:

Cerdeira-Estrada S, Martell-Dubois R, Valdéz J, Ressl R. 2019. Monthly nighttime Sea Surface Temperature Anomaly (M-NSST-A) at 1-km. Satellite-based ocean monitoring system (SATMO). Marine-Coastal Information and Analysis System (SIMAR). CONABIO. México. Dataset accessed (2018-05-10) at simar.conabio.gob.mx.

GHRSST-MUR (1jun2002-realtime): JPL MUR MEaSUREs Project. 2015. GHRSST Level 4 MUR Global Foundation Sea Surface Temperature Analysis (v4.1). Ver. 4.1. PO.DAAC, CA, USA. Dataset accessed.

Click here for additional data file.

Supplemental Information 2 Monitoring data by site and year on coral diseases from 2005 to 2019.

Susceptible species are those that presented more than 10% of SCTLD (Fig. 2; Table S2). Parque Nacional Arrecifes de Cozumel = PNAC; Parque Nacional Arrecife de Puerto Morelos = PNAPM; Biodiversity and Reef Conservation Lab, UNAM = Barco-Lab; Healthy Reefs Initiative = HRI.

Click here for additional data file.

Supplemental Information 3 Total number of colonies recorded for each species across the 82 surveyed reefs in the Mexican Caribbean (2018 and 2019).

Death colonies are only those for which death could be attributable to the SCTLD (exposed bright white skeletons; see Fig. 1).

Click here for additional data file.

We are grateful to Melina Soto, Maria del Carmen García, Alba González-Posada, Eduardo Navarro, Francisco Medellín, Blanca Quiroga, Ernesto Hevia, Eric Jordan, Nallely Hernández, Cristopher Gonzalez, Claudia Padilla, and Judith Lang who collected part of the data and/or provided insightful discussions that significantly improved the manuscript. Data from the long-term monitoring programs of the Healthy Reefs Initiative, Parque Nacional Arrecifes de Cozumel and Parque Nacional Arrecife de Puerto Morelos were essential to provide the historical background to this disease outbreak. The Comisión Nacional de Áreas Protegidas is currently coordinating the Disease Response Plan for the Mexican Caribbean, this initiative allowed us to present and discuss with a broader group of scientists, managers and NGOs our preliminary observations. The Atlantic and Gulf Rapid Reef Assessment program provided support to L.A.-F. to attend and present an earlier version of this study in the 39th meeting of the Association of Marine Laboratories of the Caribbean in Punta Cana, Dominican Republic. We thank Victor Rodríguez Ruano for their valuable time proofreading the manuscript.

Additional Information and Declarations

Competing Interests

Author Contributions

Field Study Permissions

Data Availability

The authors declare that they have no competing interests.

Lorenzo Alvarez-Filip conceived and designed the experiments, performed the experiments, analyzed the data, contributed reagents/materials/analysis tools, prepared figures and/or tables, authored or reviewed drafts of the paper, approved the final draft.

Nuria Estrada-Saldívar conceived and designed the experiments, performed the experiments, analyzed the data, contributed reagents/materials/analysis tools, prepared figures and/or tables, authored or reviewed drafts of the paper, approved the final draft.

Esmeralda Pérez-Cervantes performed the experiments, contributed reagents/materials/analysis tools, prepared figures and/or tables, authored or reviewed drafts of the paper, approved the final draft.

Ana Molina-Hernández performed the experiments, contributed reagents/materials/analysis tools, prepared figures and/or tables, authored or reviewed drafts of the paper, approved the final draft.

Francisco J. González-Barrios performed the experiments, contributed reagents/materials/analysis tools, authored or reviewed drafts of the paper, approved the final draft.

The following information was supplied relating to field study approvals (i.e., approving body and any reference numbers):

This study did not involve the collection of samples or manipulation of the habitats, therefore a permit is not needed.

We informed the Marine Protected Area (MPA) authorities prior to conducting fieldwork as it is a requirement in order to carry out activities within each MPA. The MPA authorities contacted were: Parque Nacional Arrecife de Puerto Morelos, Parque Nacional Arrecifes de Cozumel, Área de Protección de Flora y Fauna la porción norte y la franja costera oriental, terrestres y marinas de la Isla de Cozumel, Reserva de la Biósfera Sian Ka’an, Reserva de la Biósfera Banco Chinchorro, Parque Nacional Arrecifes de Xcalak and Reserva de la Biósfera Caribe Mexicano.

The following information was supplied regarding data availability:

The data used for this study is available in Table S1.

The 2018 database will be available through the Atlantic and Gulf Rapid Reef Assesment (AGRRA) program: http://www.agrra.org/.

Please note that specific accession/id numbers are not associated to datasets in Agrra. The Agrra system allows browsing for the data in a variety of different ways. To access the AGRRA database a free and open membership is requested. This is to allow Agrra to keep track of how the information is used.

See more details in http://www.agrra.org/data-explorer/.

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
