# Peer review of "A rapid spread of the stony coral tissue loss disease outbreak in the Mexican Caribbean"

_PeerJ, doi:10.7717/peerj.8069_

## Round 0.1 · original submission · Major Revisions

Thank you for your patience as we were able to secure two excellent reviewers with strong backgrounds in coral disease. Both felt the paper was strong but had a number of detailed recommendations for improvement in a revision. Once these are addressed, point by point as thoroughly as possible, we expect the paper will be acceptable for publication.

Reviewer 1 ·

Basic reporting

This manuscript documenting the spread of SCTLD is timely, important, and will add greatly to our knowledge of transmission of coral diseases. This manuscript is very well written, does an excellent job summarizing the current state of knowledge about SCTLD in the Introduction.

Experimental design

The authors follow a well established protocol for coral reef assessment and conduct appropriate statistical analysis on the data set. More details about the environmental conditions would be beneficial.

Validity of the findings

The authors have gone through the exhausting effort of surveying a great number of sites to document this disease outbreak and present a good comparison to baseline disease surveys.

This paper could be improve with a bit more data analysis, specifically more ANOSIM comparisons utilizing the additional categorical variables in the data set.

Additional comments

General Comments

Given the major negative impacts that SCTLD has wrecked upon the Florida Keys, and now the US Virgin Islands, this manuscript documenting the spread of SCTLD is timely, important, and will add greatly to our knowledge of transmission of coral diseases. The authors have gone through the exhausting effort of surveying a great number of sites to document this disease outbreak and present a good comparison to baseline disease surveys. This manuscript is very well written, does an excellent job summarizing the current state of knowledge about SCTLD in the Introduction.

This paper could be improve with a bit more data analysis, specifically more ANOSIM comparisons utilizing the additional categorical variables in the data set, see specific comment below. Additionally, it would be beneficial to the reader to have more context as to the environmental conditions of the reefs. Are some patch versus fore reefs? Are some more turbid or with high water flow or near cities. Any context into the ecology of these reefs helps but disease dynamics into better context.

Did you observe any instances of direct transmission? I know that on low coral cover reefs, it is unlikely to have two colonies close enough for direct transmission to occur. But either way (observed or not), it would be good information to include. Similarly, the caption in Figure 1 mentions the corallivore eating the diseased colony, but it is not mentioned in the text. This is a good example of potential vector-transmission that is worth mentioning, especially if you observed a fish eat from a diseased colony and then from a healthy colony. Studying and breaking transmission cycles is a clear next step in coral disease investigation, but it is hard to assess transmission if no authors comment on it at all in their manuscripts.

I recommend Accept with Minor Revisions.


Specific Comments

Keywords: Consider using SCTLD as a keyword, as this is the abbreviation being used specifically for this recent disease outbreak. I think it will help readers find your manuscript easier online.

Introduction Lines 70-73: This information is interesting, but seems out of place, especially as an ending sentence to a long paragraph about coral diseases. Consider removing this sentence, or moving it higher in the paragraph so that it fits with the chronological presentation of the other diseases.

Methods Line 122: Can you please explain the large range in replicate transects? Was this to due with size and shape of reefs. What was the average number of replicate transects per site. This will help a reader better understand the robustness of the dataset.

Methods Line 167: Consider creating a new paragraph at this line "We then infer..."

Methods Line 167-177 : The way these sentences are written makes it sound like you are comparing the physical area of the reef. Is this not the case. You are assessing the variability of the community in the context of the nMDS. These sentences (and their corresponding sentences in the Results/Discussion Lines 255-261) need to be re-written for clarity. What SEAc is doing is similar to jack-knifing, where the data set has been rarefied and re-sampled in order to get a sense of the variation and sampling depth. I don't think this needs to be explained in the results section. The ANOSIM gives the statistic about whether there are differences between groups. I would recommend giving the ANOSIM statistic and then just qualitatively describing the little overlap between groups on the nMDS plots, stating that this corresponds to compositional changes in coral reefs communities. Along these lines, I do not think that Table 1 is even necessary. I think it would be more interesting if you conducted other comparisons, such as differences between sites before the outbreak, and then assess whether communities from these sites converge onto the same nMDS space after the outbreak. Then a table with the various ANOSIM comparisons would be useful.

Results/Discussion Line 183-185: This is rather a long sentence. Consider making a new sentence starting at "this outbreak should be..."

Results/Discussion Line 191: This disease outbreak may not be linked to above normal thermal stress, but the lack of subsidence during the winter suggests that thermal connection. Can you elaborate on whether the winter sea temperatures were warmer before the outbreak and/or whether the summer temperatures were warmer than normal (although maybe not so warm as to raise the alarm for bleaching). Was there high rainfall in any of these regions before the outbreak? I think that adding some environmental data would give the prevalence data sets more context.

Results/Discussion Lines 201-205: I disagree, especially with the last sentence in this line. As you state in the Introduction that White Plague disease is also a major disease, contributing potentially more to coral declines, yet has a broad host range. Maybe not as broad as SCTLD, but still a broad host range disease is not unprecedented. There is also Black Band Disease which has also contributed greatly to coral decline, and has a very high host range. Please change These sentences to reflect the precedent of WP and BBD in he wider Caribbean. Maybe the impacts of SCTLD are unprecedented for this particilar region of the Caribbean, if so, then need context and more specific wording.

Results/Discussion: I am glad that you brought up local extinctions, because it is not often addressed in bleaching or disease outbreak papers. I hope that you are tagging these colonies to follow them over time.

Results/Discussion Line

Figure 1: Great photo

Figure 3: The caption says that the data represent the top 15 most affected species but the methods say top 11 most affected species. Please correct. Also, consider reformatting the legend so that the dots are presented in numerical order. Also consider giving context to the geography (i.e. major cities, rivers, ports etc).

Reviewer 2 ·

Basic reporting

No comment

Experimental design

No comments

Validity of the findings

A more in depth explanation of the representativity of the study, to allow generalizations is required, as expressed in the next sectioin: General comments to the author

Additional comments

An interesting and important study.

A few comments in order to improve the quality of the ms

L 93 “… affecting more than 20 species of corals (FEDP, 2019), usually in a specific order.”

Please, support this statement. While meandroid growth forms seem most susceptible, the specific order as generalize statement requires more support.

l181: “Here we describe how the SCTLD affected 82 reef sites distributed along 450 km in the Mexican …”
The surveyed sites comprise the whole reef profile or are constrained to a reef zone per site. If so, all surveys were carried out in similar reef zones?
How the 450km figure was arrived at?

l 196: “ …prevalence in the Mexican Caribbean was below 10%, reaching its lowest point in 2016-2017, just one year before the SCTLD outbreak in this region, with only 1% (Fig. 4)”.

Please consider the following:
If disease incidence is unknown, prevalence values in time are dependent on mortality through time. i.e. prevalence in t1 is not linearly related to prevalence in t2, as totals and diseased coral numbers change.

Before SCTLD, severe epizootics mostly affected specific groups (Acroproras WB, Orbicellas YBS). Therefore the overall prevalence values may be driven by a few species, not truly representative of the coral community condition.

l 202 “…however, these events have decimated the populations of only a few species (e.g. white-band and white-pox in Acropora palmata and Acropora cervicornis).”
I believe that Orbicellas have been severely impacted by YBS, as well (See Edmunds, and others), a fact that is recognized below in this ms. Consider re-phrasing

l 207: “… the following being the most affected (i.e. % disease prevalence)”
Percentage symbol is not necessary; by definition prevalence is expressed as %.

On the same tenet, prevalence is a recognized epidemiological numerical indicator, whereas affected is a colloquial expression, not related to a numerical scale.

l 207 “…Dendrogyra cylindrus (57%)”
A percentage value based on 7 colonies (fig.2) may appear dubious. Better to express it as 4 out of 7 colonies, and perhaps make some comment in fig.2

l 215 “We still found healthy colonies of … “
This statement implies that a census was carried out instead of a survey. Please modify.

l 232 “… further increase the dominance of A. agaricites and P. astreoides in the region. This may have started to become apparent already.

Regarding the last statement: These species have been becoming important long before SCTLD appearance, mostly due to disease driven coral mortality events in Caribbean reefs.


l 247 “…of the colonies were already dead as a consequence of the SCTLD “

“...were already dead probably as a consequence… “, seems more appropriate as it is a post-mortem observation.


l 261 “This is particularly explained by the sudden increase of afflicted colonies and the number of dead colonies, especially from the species Meandrina meandrites, Pseudodiploria strigosa, Diploria labyrinthiformis and Eusmilia fastigiata”.

While the statement is probably correct, the scope of it is not clear enough. For instance, Pseudodiploria strigosa and Meandrina meandrites, and to some extent Diploria labyrinthiformis tend to be common to abundant in some coral assemblages, but their relative importance vary among reef zones and relative degree of reef development. So, in principle the above statement may hold for some conditions, but not for others. To me, the wording implies that each surveyed site comprise the whole reef gradient, if so, it should be clearly stated.

I believe it is fundamental that the authors clarify this to strengthen their work. As it is, the reader wonders in what reef zones the sampling was carried out and if that allows generalizing to the whole reef. In the same tenet, it will be adequate to explain the selection criteria used to choose the 14 sites.

I believe that even if results relate to a single reef zone it does not diminish the relevance of their findings. And if that were the case, restricting the scope of the claims, make for a more credible contribution.

298 “However, the SCTLD is likely to become the most contagious and deadly coral disease ever recorded”

Please clarify in what sense is contagious used; and how that is known


Other: Several citations not included in references, among them Chan etal, 2019 and Precht, 2019

---

## Round 0.2 · accepted · Accept

Both reviewers made extensive minor comments, all of which you attended to both thoroughly and attentively, for which I am grateful. It is clear that these additions improved the clarity and impact of an already excellent paper, and we are pleased to accept it.